# Study on Kinetics of Carbonization Reaction of Hardened Cement Paste Powder Based on Carbonization Degree

**DOI:** 10.3390/ma16072584

**Published:** 2023-03-24

**Authors:** Chenhui Zhu, Yibo Jiang, Qizhi Shang, Yuchen Ye, Jie Yang

**Affiliations:** 1School of Transportation and Civil Engineering, Nantong University, Nantong 226019, China; 2Jiangsu Huaiyin Water Conservancy Construction Co., Ltd., Huaian 223000, China

**Keywords:** hardened cement paste powder, kinetics of carbonization reaction, carbonization degree, carbonization curing

## Abstract

The hardened cement paste powder (HCP) powder, devoid of the hydration cementing property, can be regenerated and cemented into a test block with practical strength of almost 60 MPa via CO_2_ carbonization using appropriate means. This study established a kinetic model of CO_2_ curing of an HCP powder test block based on the degree of carbonization to study the carbonization reaction kinetic characteristics of the test block. The model was modified according to the characteristics of the evident temperature differences in the reaction kettle in the early, middle, and late stages of the carbonization process. The proposed model can be used to formulate and control the carbonization and cementation processes of HCP powder and can also be applied to describe the kinetics of the reaction processes of other similar systems.

## 1. Introduction

The issue of global climate change is a pressing concern for humanity, and scientists around the world are actively seeking ways to mitigate or even solve this problem. One of the most direct and effective methods is to reduce the concentration of carbon dioxide in the atmosphere, as excessive CO_2_ emissions are a fundamental cause of global warming [1]. However, the cement and concrete industry is a major contributor to carbon emissions, accounting for approximately 8% of the total global carbon emissions [2,3,4]. Therefore, it is crucial to focus on reducing, capturing, storing, and utilizing CO_2_ in the cement and concrete industry. The most common research in this field involves CO_2_ reduction, which includes reducing carbon emissions during cement production and finding alternatives to cement [5,6,7].

One of the main research directions is the use of CO_2_ curing or strengthening cementitious materials. On the one hand, this can effectively capture, store, and utilize CO_2_. On the other hand, it can accelerate the development of strength in cementitious materials, shorten their curing time, and even improve their physical and mechanical properties [8,9,10,11,12,13,14,15,16,17,18,19]. The cementitious materials here include waste concrete materials such as recycled aggregate in addition to fresh cementitious materials. The basic principle is to use the carbonizable components in the cement paste attached to the surface of recycled aggregates to react with the CO_2_ gas, breaking chemical bonds and regenerating the material to improve its performance [16,20,21,22,23,24,25,26,27]. However, in addition to recycled aggregates, waste concrete also contains a large amount of crushed hardened cement paste that has not been well utilized. The authors of this article used the characteristic that CO_2_ gas can react with the components in the hardened cement paste and have a strengthening effect. They used the hardened cement paste as the raw material to prepare hardened cement paste powder test blocks and studied whether CO_2_ curing could regenerate and reuse them. The research results showed that the carbonation reaction of the hardened cement paste powder test block through CO_2_ curing can regeneratively bond into a practical strength test block, even when it completely lost its hydration and gel properties. Under a CO_2_ pressure of 0.4 MPa and carbonation for 24 h, its compressive strength can reach more than 60 MPa [28].

The CO_2_ curing of hardened cement paste powder test blocks is a complex carbonation reaction process. In order to better analyze its carbonation reaction mechanism and facilitate subsequent research, it is necessary to establish its carbonation reaction kinetics model. Currently, research on the carbonation kinetics model of concrete materials mainly focused on predicting the carbonation depth of concrete. Scholars all over the world proposed many carbonation kinetics models, which can basically be classified into three types [29,30,31,32,33]: (1) theoretical models based on diffusion theory; (2) empirical models based on carbonation experiments; and (3) models based on diffusion theory and experimental results. The mechanism of CO_2_ curing of hardened cement paste powder test blocks is similar to the carbonation curing of concrete, which is also a solid-phase reaction. Therefore, the model established in this article is also based on diffusion theory and experimental results. The difference is that in order to better study the effect of hardened cement paste powder on the capture, storage, and utilization of carbon dioxide, this article starts from the degree of carbonation of the test block, establishes a carbonation kinetics model based on diffusion theory, and based on previous experimental results, modifies the kinetics model to consider the temperature changes during the carbonation reaction.

## 2. Experimental Methods

### 2.1. Preparation of HCP

The cement used to prepare HCP was P·II 42.5 Portland cement. The cement was mixed evenly with a water-cement ratio of 0.5 and poured into 70.7 mm cubic net slurry test blocks. The test blocks were placed in a concrete curing room with a temperature of (20 ± 2) °C and relative humidity of 95% for 24 h. Then, the test blocks were broken into blocks and placed in a steam curing chamber at a temperature of 60 °C and a relative humidity of 95% for 30 days. When the hydration degree exceeded 90%, the curing was stopped, and the HCP was ground into powder. The specific surface area of the HCP was measured to be 285 m^2^/kg.

### 2.2. Preparation and Carbonization of HCP Samples

#### 2.2.1. Preparation of Test Blocks

The compression molding mold used in the test was a self-designed prismatic stainless steel mold. The area of the compression molding surface was approximately 10 cm^2^ (3.15 cm long and 3.15 cm wide). The mold diagram was available in the existing literature. After a certain amount of HCP powder was evenly mixed with water at a ratio of 0.15 water/powder, it was immediately loaded into the mold. Subsequently, the designed pressure of 20 MPa was continuously loaded on the pressure testing machine for 60 s, and the mold was then unloaded and removed to obtain the HCP test block. By controlling the quality of the mixture loaded onto the mold, the height of the final test block was maintained at approximately 3.15 cm.

#### 2.2.2. Carbonization of the Test Block

The obtained test block was placed in an environment at a temperature of (20 ± 2) °C and ambient humidity of 70% for pre-curing to a constant weight. Then, the test block was immediately placed in the carbonization kettle at the pressure of −0.09 MPa that was evacuated for carbon dioxide curing. The designed air pressure of 0.05, 0.1, 0.2, 0.3, and 0.4 MPa, and the carbonization degree was measured after 24 h.

The specific steps for carbonization are as follows: (1) Place the pre-cured test block, which has been cured to a constant weight, in the carbonization kettle, keeping a certain distance between the test blocks. (2) Close the inlet valve and use a vacuum pump to evacuate the carbonization kettle from the outlet valve to −0.09 MPa before closing the outlet valve. (3) Open the inlet valve and slowly introduce CO_2_ gas into the carbonization chamber at a rate of 0.1 MPa per minute until the design pressure is reached. Maintain the pressure stability throughout the CO_2_ curing process by keeping the inlet valve open. (4) After the designated curing time, close the inlet valve and open the outlet valve to slowly reduce the pressure in the container to one atmospheric pressure (zero gauge pressure) before removing the carbonized test block.

### 2.3. Determination of Temperature in Carbonization Cauldron

The carbonization process of the HCP test block is an exothermic reaction, and water is one of the reaction products. The change of temperature in the carbonization kettle during the curing process partly reflects the reaction rate and degree in the curing process; therefore, the temperature was monitored during the curing process in this test. The temperature and humidity detector model 179-DTH (accuracy: ±0.4 °C) manufactured by Apresys was placed in the carbonization kettle, and the temperature data in the carbonization kettle was recorded every 1 min.

### 2.4. Determination of Degree of Carbonation

In this study, the carbonization degree of the HCP test block after carbonization was determined using the burning method. The carbonization test block was broken into particles of less than 5 mm in size and ground until all could be passed through a sieve with a pore size of 0.08 mm. Ground powder samples of 2 g were first cauterized at 500 °C for 1 h and were then cooled. The mass of the cauterized material m_500_ was weighed, and then placed in the furnace to cauterize it at 900 °C for 1 h. The cauterized material m_900_ was cooled and weighed. The calculation formula to obtain the CO_2_ content (COC) of the sample is as follows:(1)$$COC(\%)=\frac{m_{500}-m_{900}}{m_{900}}\times 100\%$$

The test was repeated three times, and the average value was considered the CO_2_ content of the HCP block. The CO_2_ content of uncarbonized HCP, fully carbonized HCP, and each carbonized sample is denoted as COC_0_, COC_∞_ (38.97% in this test), and COC_t_, respectively. The carbonization degree α of the HCP sample can be calculated according to Equation (2).
(2)$$\alpha (\%)=\frac{{\mathrm{COC}}_{t}-{\mathrm{COC}}_{0}}{{\mathrm{COC}}_{\infty }-{\mathrm{COC}}_{0}}\times 100\%$$

## 3. Results and Discussion

### 3.1. Change of Temperature during Carbonization Reaction

Figure 1 shows the temperature curve of HCP test block carbonization in the kettle for 24 h under varying CO_2_ pressure. As shown in the figure, when the CO_2_ pressure was maintained constant, the temperature in the carbonization kettle began to rise rapidly after CO_2_ gas was added, and all of them reached the highest value in approximately 20–25 min. As carbonization proceeded, the temperature in the kettle was gradually reduced to the room temperature. This indicates that the carbonization reaction rate in the specimens at the early stage of carbonization was rapid, and a large amount of heat was released, leading to the rapid rise of the temperature in the kettle. Then, the carbonization reaction rate gradually decreased, and the heat in the kettle was gradually dispersed through the kettle body. The temperature in the kettle was higher for the first 1 h of carbonization, indicating that the carbonization reaction rate is faster during this period.

With the increase in CO_2_ pressure during carbonization, the maximum temperature that can be reached in the corresponding carbonization tank also increased. When the CO_2_ pressure was 0.05, 0.1, 0.2, 0.3, and 0.4 MPa, the highest temperatures in the carbonization kettle were 22.22, 22.43, 22.95, 23.03, and 24.05 °C, respectively. This indicates that the higher the CO_2_ pressure, the faster the carbonization reaction rate and higher the heat released in the early stage of carbonization.

### 3.2. Change Law of Carbonization Degree with Carbonization Time

Figure 2 shows the variation of carbonization degree of the HCP test block with carbonization time under varying CO_2_ pressure. The figure shows that the carbonation degree of the test block under varying CO_2_ pressure had a very similar trend to that with time. The carbonization process of the sample increased with increasing carbonization time. Furthermore, the higher the CO_2_ pressure, the higher the carbonization degree of the test block after the same time; however, with the carbonization process, this gap gradually decreased. The carbonization degree of 0.4 MPa CO_2_ carbonization for 24 h reaches 80.99%. Therefore, it can be inferred that for every old concrete hardening cement powder formed by 1 ton of P·II 42.5 cement, 293.6 kg of CO_2_ gas can be absorbed and solidified to form carbonation products such as CaCO_3_ with high stability. Therefore, the HCP carbonation cementation process had high CO_2_ neutralization efficiency.

Analysis of the temperature changes inside the carbonization kettle during the carbonization process showed that the carbonization degree of the test block increased rapidly after CO_2_ was introduced into the carbonization kettle. For example, at a CO_2_ pressure of 0.2 MPa, the carbonization degree of the carbonized test block reached 39.7% after 0.25 h and reached 57.8% after 1 h of carbonization. This is because the original porosity and pore size of the test block are large, and the powder particles have a large specific surface area. The high-pressure CO_2_ can quickly diffuse into the interior of the test block and react with various components in the powder particles.

As shown in Figure 2, the carbonization rate of the specimen began to slow down after about 1 h of carbonization, and reached 62.9% after 6 h and 73.5% after 12 h. The decrease in carbonization rate was due to the increased resistance to CO_2_ diffusion through the carbonization layer of the powder particles after the initial rapid carbonization. The formed carbonization products such as CaCO_3_ not only block the capillary pores on the surface of the powder particles, reducing the diffusion coefficient of CO_2_ in the powder particles, but also precipitate in the pores between the powder particles, blocking the passage of CO_2_ from the surface of the specimen to the surface of the powder particles, resulting in a decrease in CO_2_ concentration on the surface of the powder particles. In addition, the carbonization process is accompanied by overall volume shrinkage of the specimen, which reduces the pore size and increases the diffusion resistance of CO_2_. After 12 h of carbonization, the increase in carbonization degree of the specimen became very slow. The carbonization degree of the specimen was only 3.9% higher at 24 h than at 12 h, reaching 78.4%.

## 4. Kinetic Analysis of Carbonization of the HCP Test Block

### 4.1. Carbonization Kinetics Model Based on Carbonization Degree

Assuming that the HCP powder particles are spherical, the particle morphology before and after carbonization for t hours is shown in Figure 3, where r_0_ is the initial diameter of the HCP powder particles, and r_t_ is the diameter of the remaining HCP powder particles after carbonization for t h.

Assuming that the thickness of the HCP powder particles involved in the carbonization reaction is y after t h of carbonization, as shown in Figure 1,
(3)$$y=r_{0}-r_{t}$$

Based on the reaction kinetic equation of plate diffusion model,
(4)$$\frac{dy}{dt}=\frac{k_{0}D_{1}C}{y}$$
where $\frac{dy}{dt}$ represents the diffusion rate of CO_2_ through the product layer and corresponds to the reaction rate in the plate diffusion model; D_1_ is the diffusion coefficient of CO_2_ gas in the product layer; k_0_ is the proportionality coefficient; C is the concentration of reactants on the surface of the powder particles.

In this study, the diffusion coefficient of D_1_, namely CO_2_ gas, in the product layer was constant. The CO_2_ gas concentration C_t_ ≈ C_0_ (initial concentration) on the surface of the HCP powder at the initial reaction stage is shown in Figure 4a. As the carbonization progressed, as shown in Figure 4b, the pores between the particles were gradually filled with products such as calcium carbonate, and the diffusion rate of CO_2_ from the surface of the molding body to the powder particles decreased, resulting in a decrease in the CO_2_ concentration C_t_ on the surface of the powder particles.

The filling degree of pores between particles was proportional to the carbonization degree of particles, and the CO_2_ concentration on the surface of the powder particles was inversely proportional to the carbonization degree of particles. For flat concrete samples, the carbonation depth (corresponding to the carbonation degree) was proportional to t^1/2^, that is, the diffusion resistance of CO_2_ from the surface of the molding body to the surface of the powder particles was proportional to t^1/2^. Assuming that the carbonization occurs simultaneously, the carbonization products reduce the diffusion rate of CO_2_ from the surface of the molding body to the powder particles. Then, the CO_2_ concentration C_t_ on the surface of the hardened cement paste powder particles after carbonization time t can be approximated as
(5)$$C_{t}=\frac{C_{0}}{{\mathrm{at}}^{1/2}}$$
where at^1/2^ > 1 and a is the proportionality constant. Substituting Equation (5) into Equation (4), we obtain
(6)$$\frac{dy}{dt}=\frac{k_{0}DC_{0}{/(\mathrm{at}}^{1/2})}{y}=\frac{k_{1}}{{\mathrm{yt}}^{1/2}}$$
where k_1_ is the proportional coefficient, which can be obtained by integrating Equation (6):(7)$$y^{2}=k_{2}\sqrt{t}+B_{0}$$
where k_2_ is the proportionality coefficient and B_0_ is the constant. Assuming that V is the remaining volume of the HCP powder particles at time t,
(8)$$V=\frac{4\pi }{3}r_{t}^{3}$$

In this case, let α be the degree of carbonation at time t. In this test, the degree of carbonization refers to the ratio of the amount of CO_2_ consumed in the reaction to the theoretical maximum carbon dioxide consumption, that is, the ratio of the volume of the HCP powder in the carbonization reaction to the original volume of the particle:(9)$$\alpha =\frac{V_{r_{0}}-V_{r_{t}}}{V_{r_{0}}}=\frac{\frac{4\pi }{3}r_{0}^{3}-\frac{4\pi }{3}r_{t}^{3}}{\frac{4\pi }{3}r_{0}^{3}}=1-(\frac{r_{t}}{r_{0}}{)}^{3}$$

From Equations (8) and (9), we obtain
(10)$$V=\frac{4\pi }{3}r_{0}^{3}(1-\alpha )$$

By combining Equations (8) and (10),
(11)$$\frac{4\pi }{3}r_{0}^{3}(1-\alpha )=V=\frac{4\pi }{3}r_{t}^{3}\to r_{{}^{t}}=r_{0}{\left(1-\alpha \right)}^{\frac{1}{3}}$$

Substituting Equation (3) into Equation (11),
(12)$$y=r_{0}\left[1-{\left(1-\alpha \right)}^{1/3}\right]$$

Finally, Equations (7) and (12) are simultaneously established to obtain a preliminary kinetic model:(13)$${\left[1-(1-\alpha {)}^{1/3}\right]}^{2}=\frac{k_{2}\sqrt{t}+B_{0}}{r_{0}{}^{2}}=k\sqrt{t}+B$$
where k is the reaction coefficient of the entire reaction and B is a constant.

### 4.2. Fitting Analysis of Test Results and Kinetic Model

#### 4.2.1. Analysis of Fitting Results

To test the accuracy of the relationship between the curing degree and curing time of CO_2_, regression analysis was carried out on the established model using experimental data. The carbonization degree data of the test block after different carbonization times in Figure 2 were substituted into Equation (13) for linear regression, and the regression analysis results are shown in Figure 5, where are linear regression analysis graphs of the carbonization degree and carbonization time of the test block under CO_2_ pressure of 0.05, 0.1, 0.2, 0.3, and 0.4 MPa, respectively.

When the CO_2_ pressure was 0.05 and 0.1 MPa, the R^2^ value after linear regression was significantly higher than that under 0.2, 0.3, and 0.4 MPa, and with the increase in CO_2_ pressure, the R^2^ value decreased, indicating that with the increase in CO_2_ pressure, Equation (13) became inconsistent with the actual reaction process. In addition, intercept B in Equation (13) was not 0, indicating that the reaction degree α was not 0 when t was 0, which was evidently inconsistent with the reality.

#### 4.2.2. Effect of CO_2_ Pressure on k and B Values

Equation (13) is a carbonization kinetic model based on carbonization time and degree, which can be used to predict the relationship between carbonization time and degree of the test block. The reaction rate constant k in Equation (13) is related to the CO_2_ concentration on the surface of the test block, that is, the CO_2_ pressure. The higher the CO_2_ pressure (concentration) in the carbonization kettle, the greater the k value.

It can be observed from Table 1 that with the increase in CO_2_ pressure, the values of k and B in the formula also increased, and the relationship is shown in Figure 6. As shown in the figure, in the relation between the reaction rate constant k and the CO_2_ pressure pco_2_, k was not 0 when pco_2_ was 0, which was inconsistent with the reality.

### 4.3. Modification of the Kinetic Model of Carbonation

Figure 5 shows that there was a turning point in the measured data when the carbonization time was approximately 1 h, and the linear relationship of the measured data after 1 h was good. It can be observed that there were slight differences in the reaction mechanism in the early, middle, and late stages of the carbonization reaction.

Equation (4) is established under certain assumptions, and therefore, the carbonation process should be divided into two periods of 0–1 h and 1–24 h for analysis when kinetic model regression is carried out. When the CO_2_ concentration C_t_ ≈ C_0_ (initial concentration) on the surface of the HCP powder particles during 0–1 h of carbonization is considered, $\frac{dy}{dt}=\frac{k_{0}D_{1}C_{0}}{y}$ is the diffusion rate of CO_2_ gas through the product layer during this period. D_1_ and C_0_ are constants and can be obtained via integration.
(14)$$y^{2}=k_{3}t+B_{1}$$
where k_3_ is the proportionality coefficient and B_1_ is a constant. When carbonization time t = 0 and y = 0, B_1_ = 0 is substituted in Equation (14). When Equations (3)–(14) and (3)–(12) are simultaneous, we obtain
(15)$${\left[1-(1-\alpha {)}^{1/3}\right]}^{2}=kt$$
where k is the scale coefficient.

Linear regression analysis was performed on the measured carbonization degree data of the sample under different CO_2_ pressures from 0 to 1 h according to equation (15), and linear regression analysis was performed on the measured carbonization degree data from 1 to 24 h according to Equation (13). Figure 7 shows the linear regression analysis diagram, where (a), (c), (e), (g), and (i) are regression analysis figures of 0–1 h under CO_2_ pressure of 0.05, 0.1, 0.2, 0.3, and 0.4 MPa, respectively; (b), (d), (f), (h), and (j) are regression analysis figures of 1–24 h under CO_2_ pressure of 0.05, 0.1, 0.2, 0.3, and 0.4 MPa, respectively.

Table 2 shows the regression analysis results under different CO_2_ pressures. Combined with the analysis in Figure 7, it can be observed that the empirical formula obtained according to Equation (13) was consistent with the measured data from 1 to 24 h, and all the R2 values were close to 1, indicating that the measured data in this period was well-correlated with the kinetic Equation (13).

However, during 0–1 h, the measured data under the CO_2_ pressure of 0.05 and 0.1 MPa had a good correlation with the fitted curve obtained according to Equation (15). With increasing CO_2_ pressure, the deviation degree between the test results and the Yangder equation ([1 − (1 − G)^1/3^]^2^ = kt) slightly increased. This was because for a higher CO_2_ pressure, the carbonation degree of the test block develops faster, the pores between the particles are filled faster, and the CO_2_ concentration difference between the powder particle surface and the test block surface is affected earlier. Therefore, under the experimental conditions in this study, the carbonization degree α of the HCP powder conformed to the following kinetic model:$$\begin{array}{l} (1)\;\mathrm{When}\;0\;\leq \;t\;<\;t_{1},\;F_{J}(\alpha )=\mathrm{kt};\\ (2)\;\mathrm{When}\;t_{1}\;\leq \;t\;\leq \;24,\;F_{J}(\alpha )=k\sqrt{t}+B; \end{array}$$
where F_J_(α) is the reaction rate (degree of carbonization) function ($F_{J}(\alpha )={\left[1{-(1-\alpha )}^{1/3}\right]}^{2}$) and t_1_ ≈ 1 h under the experimental conditions in this study.

The HCP powder test block has the characteristics of high porosity and large pore size. At the initial carbonization stage (0 ≤ t < t_1_), the CO_2_ gas on the surface of the test block can quickly enter the inside of the test block and reach the surface of the powder through the alkali pores of the HCP powder particles. The CO_2_ concentration on the particle surface is not different from that on the surface of the test block. The kinetics of the carbonization process of the powder particles conforms to the Yangder equation:(16)$${\left[1-(1-\alpha {)}^{1/3}\right]}^{2}=kt$$
when the carbonation reaction reaches the middle and late (t_1_ ≤ t ≤ 24) stages, the powder intergranular pore with carbide products has a blocked part and the resistance to the spread of CO_2_ gas from the block surface increases gradually. On the surface of the powder particles the CO_2_ concentration and the concentration difference on the surface of the specimen increase, and therefore, the carbonization process of powder particle dynamics is in conformity with the modified Yangder equation:(17)$${\left[1-(1-\alpha {)}^{1/3}\right]}^{2}=k\sqrt{t}+B$$

## 5. Conclusions

In previous research, the authors of this paper found that using CO_2_ to cure HCP specimens can improve their physical and mechanical properties, while also solidifying a certain amount of CO_2_ gas, achieving multiple benefits. Therefore, to better study this carbonation process and to better understand the solidification effect of CO_2_ in the process, this paper established a carbonation kinetics model based on the diffusion theory and the degree of carbonation of the specimens. Combining the temperature changes during the carbonation reaction, the carbonation kinetics model was corrected.

The research results showed that within 1 h of carbonation, due to the relatively large original porosity and pore size of the specimens and the large specific surface area of the powder particles, CO_2_ at relatively high pressure can quickly diffuse into the interior of the specimens and react with various components of the powder particles. Therefore, the carbonation reaction rate of the specimens was fast, and a large amount of energy was released, resulting in a rapid increase in the degree of carbonation. During this period, the carbonation kinetics of the HCP specimens complied with the Jander equation. After 1 h of carbonation, however, due to the product layer covering the surface of the HCP particles and the decrease in the concentration of CO_2_ around the deep particles, the carbonation reaction rate of the specimens in this period was slow, the energy released was small, and the degree of carbonation increased slowly. During this period, the carbonation kinetics complied with the Jander equation revised in this paper.

## Figures and Tables

**Figure 1 materials-16-02584-f001:**
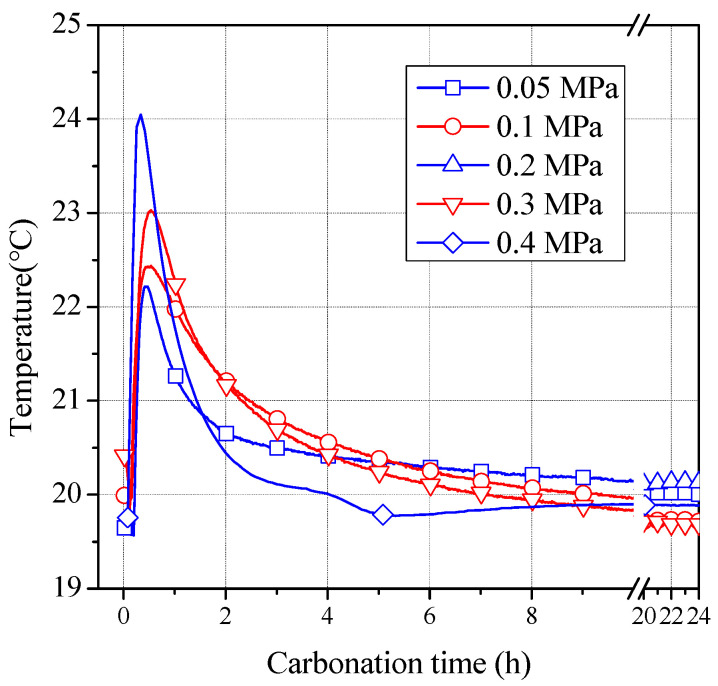
Temperature profiles of the carbonization kettle carbonated under varying CO_2_ pressure.

**Figure 2 materials-16-02584-f002:**
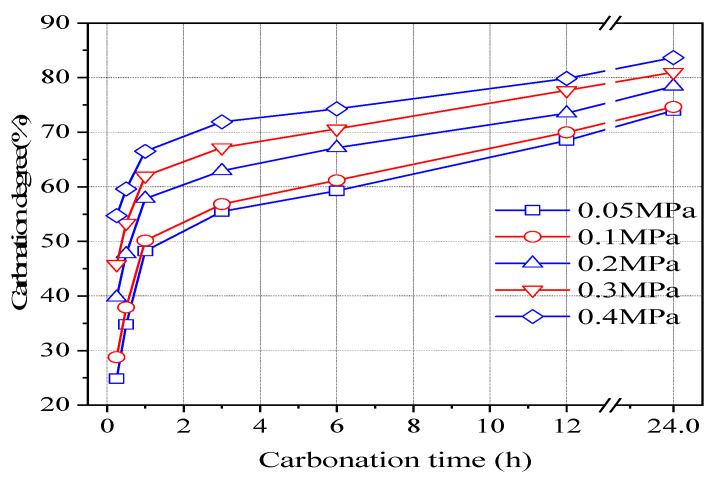
Carbonation degree of compacts for different CO_2_ pressures vs. carbonation time [16].

**Figure 3 materials-16-02584-f003:**
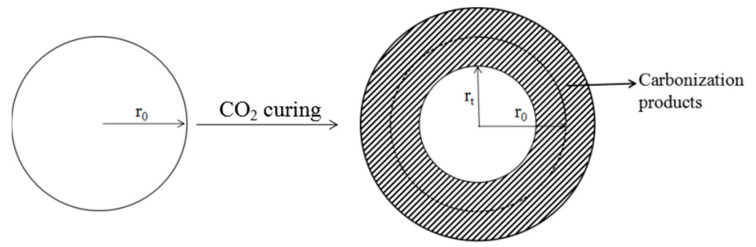
Schematic of the particles in compacts before and after CO_2_ curing.

**Figure 4 materials-16-02584-f004:**
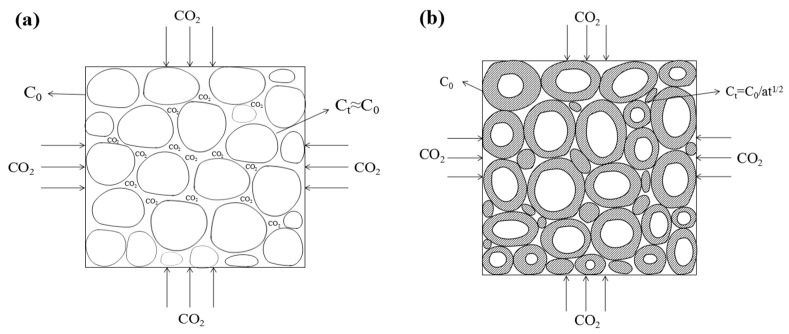
Schematic diagram of CO_2_ concentration on the particle surface. ((**a**) shows the initial stage of carbonization, (**b**) shows the carbonization process).

**Figure 5 materials-16-02584-f005:**
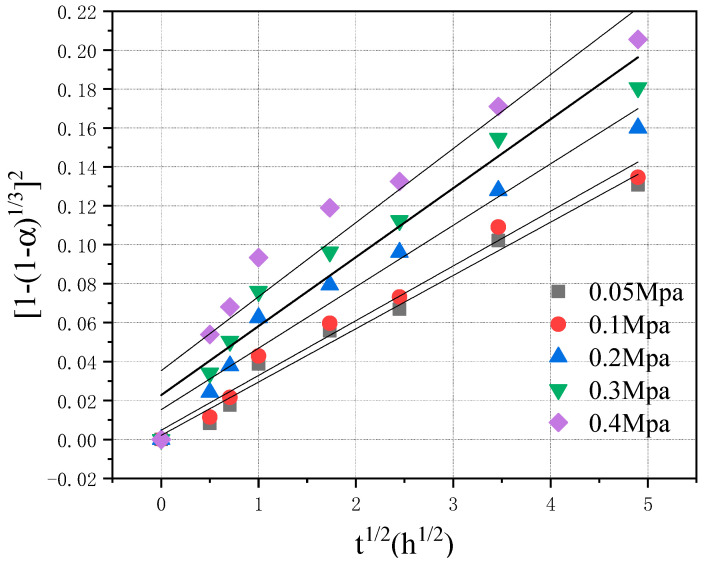
Test results and kinetic model of regression analysis.

**Figure 6 materials-16-02584-f006:**
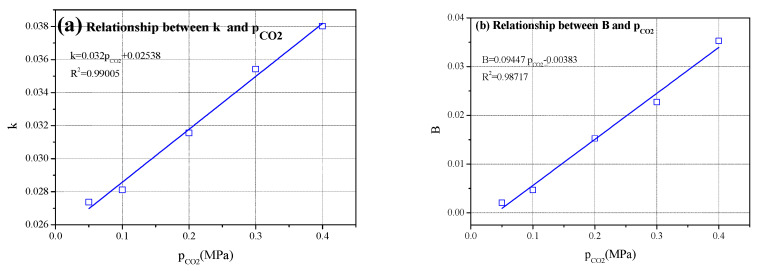
Relationships between k and B with CO_2_ pressure.

**Figure 7 materials-16-02584-f007:**
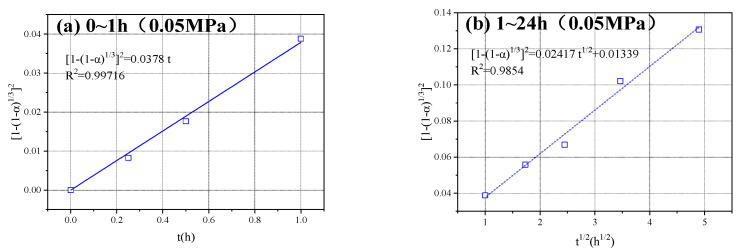
Linear-regression analysis under varying CO_2_ pressure.

**Table 1 materials-16-02584-t001:** Parameters of linear-regression analysis.

CO_2_ Pressure/MPa	y = kx + B		Standard Error		R^2^
	Slope k	Intercept B	k	B	
0.05	0.02737	0.00207	0.0015	0.00364	0.97922
0.1	0.02812	0.00472	0.00169	0.00409	0.97522
0.2	0.03156	0.01526	0.00251	0.00606	0.95743
0.3	0.03542	0.02274	0.00342	0.00826	0.93834
0.4	0.03802	0.03528	0.00442	0.01068	0.91264

**Table 2 materials-16-02584-t002:** Results of the linear-regression analysis under different CO_2_ pressures.

Carbonization Pressure /MPa	0–1 h		1–24 h	
	$F_{J}(\alpha )=\mathrm{kt}$	R^2^	$F_{J}(\alpha )={\mathrm{kt}}^{1/2}+B$	R^2^
0.05	$F_{J}(\alpha )=0.0378t$	0.99716	$F_{J}(\alpha )=0.02417t^{1/2}+0.01339$	0.98540
0.1	$F_{J}(\alpha )=0.04308t$	0.99972	$F_{J}(\alpha )=0.02436t^{1/2}+0.01792$	0.98252
0.2	$F_{J}(\alpha )=0.06667t$	0.97926	$F_{J}(\alpha )=0.02552t^{1/2}+0.03602$	0.99548
0.3	$F_{J}(\alpha )=0.08361t$	0.95968	$F_{J}(\alpha )=0.02785t^{1/2}+0.04853$	0.97626
0.4	$F_{J}(\alpha )=0.10728t$	0.90693	$F_{J}(\alpha )=0.02891t^{1/2}+0.06592$	0.99016

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
