# Peer review of "Study on Kinetics of Carbonization Reaction of Hardened Cement Paste Powder Based on Carbonization Degree"

_materials, 2023, doi:10.3390/ma16072584_

Round 1

Reviewer 1 Report

Though the proposed topic is of interest for the journal audience, the paper lacks in-depth scientific explanations. The paper is just a scratch from the observations. Hence, the paper requires a significant improvement before the reviewer makes a suggestion for the proposed work.

1. Introduction requires a major improvement. Only limited literature works are included which is not sufficient to justify the novelty of the proposed work.

2. The quantum of work performed is very limited to publish in a standard journal. Hence, the authors should further improve the quantum of work to be included as a part of this submission.

3. Table 1 and 2 are already well-studied in the past. Hence, it is to be removed from the revised submission.

4. Elaborate the standard procedure used for performing carbonation test.

5. Discussions pertaining to the experimental observations are too weak and limited. This section is to be significantly improved in the revised submission.

6. Conclusions should sound like a real conclusion and not like a mere summary. Please improve.

Author Response

Dear reviewer,

Thank you very much for your valuable suggestion and for pointing out the shortcomings in the article. I have made the necessary modifications based on your suggestions. If there are any other areas that require further improvement, please do not hesitate to let me know. The detailed corrections are listed below point by point:

  1. Introduction requires a major improvement. Only limited literature works are included which is not sufficient to justify the novelty of the proposed work.

Response 1: Regarding the introduction section of the article, I sincerely inform you that I have carefully revised it and provided a clear explanation of the significance and importance of this work, as well as the differences between this study and existing research.

  1. The quantum of work performed is very limited to publish in a standard journal. Hence, the authors should further improve the quantum of work to be included as a part of this submission.

Response 2: As you have pointed out, this study is built on previous research, and therefore, there are certain limitations. However, in future research, I will explore this area more extensively and try to do more research in this area.

  1. Table 1 and 2 are already well-studied in the past. Hence, it is to be removed from the revised submission.

Response 3: Thank you for your advice, and I have deleted Tables 1 and 2 from the revised submission.

  1. Elaborate the standard procedure used for performing carbonation test.

Response 4: In the revised manuscript, I have provided a detailed explanation of the standard procedure for the carbonization experiment. The main reason for not providing a detailed explanation in the original manuscript was that this experimental procedure had already been described in the reference "Carbonation-cementation of recycled hardened cement paste powder". Thank you for your understanding.

  1. Discussions pertaining to the experimental observations are too weak and limited. This section is to be significantly improved in the revised submission.

Response 5: I have added a discussion on the experimental results in the revised manuscript, and your suggestions have been extremely helpful in making these modifications. Thank you once again for your valuable feedback.

  1. Conclusions should sound like a real conclusion and not like a mere summary. Please improve.

Response 6: Finally, under your guidance, I have improved the conclusion section of the manuscript.

Thank you again for your invaluable feedback, and I look forward to receiving your continued guidance in future work.

Best regards,

[Chen-Hui Zhu]

Reviewer 2 Report

The authors have provided a kinetic analysis of the carbonization reaction of hardened cement paste powder in a fairly detailed and rational manner, however the following comments should be taken into account:

In the introductory part, the significance of this work for the professional and scientific community is not very clear.

2.1. The granulation of the material (cement and HCP) should be shown by the authors (important parameter for kinetic analysis, eg., Fig 4).  the effect of pores on kinetics is mentioned later

2.2.2. the pressure of -0.09 MPa (negative, relative to atm? or this is just a dash)

Fig. 5. The authors used equation 13 as a kinetic model - according to figure 5 there is not much data agreement with this model (it appears that R2 is relatively low) - this is inconsistent with table 3 (R is relatively high). It can be seen that by increasing the pressure the correlation is lost. During 0–1 h, by increasing the CO2 pressure, the correlation with the selected kinetic model is lost; the authors have provided an explanation for this - it may be acceptable. However, perhaps at higher pressures some other mechanisms can also occur simultaneously (due to chemical reactions), also here the pore size of the initial material (or perhaps granulation) is important. It would be good if the authors had a method to confirm their statement about the pore size during the process.

The authors approximated that the total chemical reaction is reduced and generalized to CaO + CO2 = CaCO3; however, when we consider the composition, there can be complex mechanisms (especially with Si and Mg oxides) that can affect the kinetics. This needs to be discussed.

Equations are repeated in the concluding part, perhaps not relevant.

Author Response

Dear reviewer,

Firstly, thank you for providing valuable feedback and pointing out the shortcomings of the article. I have made revisions to the article based on your suggestions, and I would appreciate it if you could provide further guidance if any changes are still needed. Below are my responses to your comments:

  1. In the introductory part, the significance of this work for the professional and scientific community is not very clear.

Response 1: The author has carefully revised the introduction section of the article to highlight the significance and importance of our work in the professional and scientific community, as well as the differences from previous research.

  1. The granulation of the material (cement and HCP) should be shown by the authors (important parameter for kinetic analysis, eg., Fig 4). the effect of pores on kinetics is mentioned later.

Response 2: This work is based on previous research, and we have already published a paper on the effects of particle shape and pore structure on the carbonization process in the literature "Carbonation-cementation of recycled hardened cement paste powder". We did not provide a detailed description in this article, and we apologize for this. I have attached the relevant content of the article, thank you for pointing out the problem in this aspect。

  1. the pressure of -0.09 MPa (negative, relative to atm? or this is just a dash)

Response 3: Yes, the symbol refers to negative pressure, which means a state of pressure lower than atm.

  1. Fig. 5. The authors used equation 13 as a kinetic model - according to figure 5 there is not much data agreement with this model (it appears that R2 is relatively low) - this is inconsistent with table 3 (R is relatively high). It can be seen that by increasing the pressure the correlation is lost. During 0–1 h, by increasing the CO2 pressure, the correlation with the selected kinetic model is lost; the authors have provided an explanation for this - it may be acceptable. However, perhaps at higher pressures some other mechanisms can also occur simultaneously (due to chemical reactions), also here the pore size of the initial material (or perhaps granulation) is important. It would be good if the authors had a method to confirm their statement about the pore size during the process.

Response 4: Thank you for your valuable comments. The reason for not using higher pressure is mainly due to equipment limitations, which also led to the limitations of the kinetic model presented in this article. In addition, the pore structure of the materials and the changes in the carbonization process have been explained in the literature "Carbonation-cementation of recycled hardened cement paste powder". Therefore, the effects of pore structure on the carbonization process have been considered in our analysis, and we apologize for not explaining it in detail in the article.

  1. The authors approximated that the total chemical reaction is reduced and generalized to CaO + CO2= CaCO3; however, when we consider the composition, there can be complex mechanisms (especially with Si and Mg oxides) that can affect the kinetics. This needs to be discussed.

Response 5: We apologize to the reviewer for not providing a detailed explanation of the chemical composition changes in the sample during the carbonization process. Our previous study revealed that the main product of carbonization curing is calcite, with the main component being CaCO3. Therefore, we simplified the overall chemical reaction as CaO + CO2 = CaCO3.

  1. Equations are repeated in the concluding part, perhaps not relevant.

Response 6: With your guidance, we have further improved the conclusion in the revised manuscript.

Thank you again for your invaluable feedback, and I look forward to receiving your continued guidance in future work.

Best regards,

[Chen-Hui Zhu]

Round 2

Reviewer 1 Report

Most of the comments were addressed by the authors. However, they are elaborate and exhaustive.